

# Top-k sentiment analysis over spatio-temporal data

Abdulaziz Almaslukh, Aisha Almaalwy, Nasser Allheeib, Abdulaziz Alajaji, Mohammed Almukaynizi and Yazeed Alabdulkarim

Department of Information Systems, College of Computer and Information Sciences, King Saud University, Riyadh, Saudi Arabia

## ABSTRACT

In recent years, social media has become much more popular to use to express people's feelings in different forms. Social media such as X (*i.e.*, Twitter) provides a huge amount of data to be analyzed by using sentiment analysis tools to examine the sentiment of people in an understandable way. Many works study sentiment analysis by taking in consideration the spatial and temporal dimensions to provide the most precise analysis of these data and to better understand people's opinions. But there is a need to facilitate and speed up the searching process to allow the user to find the sentiment analysis of recent top-k tweets in a specified location including the temporal aspect. This work comes with the aim of providing a general framework of data indexing and search query to simplify the search process and to get the results in an efficient way. The proposed query extends the fundamental spatial distance query, commonly used in spatial-temporal data analysis. This query, coupled with sentiment analysis, operates on an indexed dataset, classifying temporal data as positive, negative, or neutral. The proposed query demonstrates over a tenfold improvement in query time compared to the baseline index with various parameters such as top-k, query distance, and the number of query keywords.

## INTRODUCTION

Modern enterprises typically receive extensive amounts of data in increasing fashion. This data is often stored in a final data warehouse for analytical purposes (*Taniar & Rahayu, 2022*). It can be used for querying the daily operation metrics, building various dashboards that support the business needs, and often can be utilized to build predictive models. Processing this data can be challenging and time consuming if the data infrastructure has not been designed carefully. The data is normally huge in size and arriving at a rapid rate, and often comes in different forms structured and unstructured such as social media, satellite, and Internet of things (IoT) data.

Sentiment analysis is considered as one of the main building blocks of natural language processing (NLP) techniques that is used intensively to extract the opinions of user-generated textual data that are posted in various online platforms such as X platform (*Poria et al., 2020*; *Alfarrarjeh et al., 2017*). This analysis classifies the opinions as positive, negative, or neutral (*Parimala et al., 2021*) and some techniques are score based rather than a class. Sentiment analysis can be found in different sectors such as businesses,

Corresponding author
Abdulaziz Almaslukh,
aalmaslukh@ksu.edu.sa

education, public health, transportation, disasters, governments (*Chaturvedi, Toshniwal & Parida, 2019*; *Alves et al., 2015*; *Shah et al., 2019*; *Parimala et al., 2021*). This analysis helps decision-makers in enhancing their strategies and responses and in monitoring behaviors that could significantly impact the organization.

Geo-search queries have received significant attention from the research community due to the applicability in various critical domains such as urban planning (*Hristova et al., 2016*), rescue missions (*Mehta, 2017*; *MacMillan, 2017*; *Rhodan, 2017*), and disease tracking and prevention. Over the last two decades, several variations of geo-search queries have been proposed in the literature such as social queries (*Ahuja et al., 2015*; *Almaslukh, Kang & Magdy, 2021*), temporal queries (*Magdy et al., 2014a*), keyword queries (*Chen, Cong & Cao, 2013a*), over snapshot data (*Gutiérrez et al., 2005*), and over streaming data environment (*Almaslukh & Magdy, 2018*). A major class of these queries that has been applied extensively in real word applications is the geo-keyword temporal queries (*Hoang-Vu, Vo & Freire, 2016*) that focus on three dimensions: space, time, and keywords. These queries return the data that satisfy the three predicates. Since the result of the queries is normally huge in size, top-$k$ is used to limit the result based on one of the three predicates or combine the three together based on a given ranking function. For instance, "find top 10,000 tweets mentioning the ChatGPT model keywords posted recently in Tokyo". Particularly we focus on spatial range queries, coined by *Taniar & Rahayu (2015)*. Spatial range or distance queries involve identifying objects within a specific range or radius, encompassing a broad range of tasks, including both finding objects of interest and defining the range or region itself.

While sentiment analysis is useful to analyze the data without taking into account other dimensions, it can be even more useful if the spatial and temporal dimensions have been taken into account while performing the analysis. It can provide more focused analysis to better understand the user's opinions in different locations at different time intervals. Various arrays of analytical queries need to explore the user-generated data with the spatial and the temporal dimensions in addition to a specific topic. These dimensions could be challenging and complicated especially if the underlying applications are critical and cannot tolerate significant latency. Existing techniques suffer from processing this query efficiently as these techniques do not support the sentiment analysis while taking into account the textual, spatial, and temporal dimensions. As a result, the query time can be unacceptable especially for real-time applications such as stocks and cryptocurrencies analysis.

To address this issue, this work proposes a new analytical query over user-generated data named *GeoSentiment* to efficiently process the sentiment analysis while incorporating the space and time of the data. This query can be utilized in various problem settings in order to help the enterprise process their accumulated data effectively, respond to a potential risk more quickly, and can be a building block for more rigorous analytical queries. More specifically, the input geo-data is analyzed by using one of the NLP techniques. Then, the data fed into a hybrid index which considered the textual and temporal dimensions in addition to the spatial while the sentiment analysis score is embedded. To process the proposed query efficiently, we develop a processor that takes advantage of the constructed index to smartly prune irrelevant data and process data that

contributes to the final output. The query returns the final result as sentiment scores output with respect to the query inputs including the topic.

The spatial distance query is the focus of this work where the result of the query is the overall sentiment analysis score for the set of top-$k$ geo-objects each of which satisfy the query predicates including the keyword, time, and the given region. The experiment results show a significant improvement by using the hybrid index structure over the baseline index which only indexes the spatial aspect without considering the object keywords. Utilizing the hybrid index reduces the query time by one magnitude. This improvement mainly derived from underlying hybrid index structure in addition to the pruning techniques that the query process utilizes while processing the data.

The main contributions of this article are summarized as follow:

- We propose scalable sentiment analysis search query that processes data objects based on spatial, temporal, and keyword predicates on pre-analyzed data.
- We propose two indexes (basic and hybrid) to answer the proposed query efficiently.
- We develop a query processor that smartly prunes the irrelevant data objects by utilizing the hybrid index structure contents.
- We evaluate the proposed query using a real Twitter dataset and compare the result with the baseline index structure.

The rest of this article is organized as follows. The "Related Work" section presents the related work with respect to our problem. The "Problem Statement" section defines the problem. The "Index Structure" and "Query Processing" sections detail the proposed sentiment indexing structure and query processing techniques. The "Experimental Evaluation" section provides an extensive experimental evaluation. Finally, the "Conclusion" section concludes the article.

## RELATED WORK

Geo-social queries have gained increased attention among researchers due to the proliferation of handheld technology (*Sohail, Cheema & Taniar, 2018*). In *Cao et al. (2012)*, the authors conducted a study on keywords and introduced a novel query type. In keyword queries, a user's query retrieves k objects that contain a specific keyword. The score of an object is computed using a function that combines the object's distance from the query and the relevance of the object's textual description with the query keywords. Spatial keyword queries have been extensively explored in Euclidean space (*Armenatzoglou, Ahuja & Papadias, 2015*; *Chen et al., 2013b*; *Cong, Jensen & Wu, 2009*; *Wu, Cong & Jensen, 2012*; *Zhang, Chan & Tan, 2014*), where the Euclidean distance serves as the metric for spatial proximity when calculating spatio-textual scores. *Samet, Sankaranarayanan & Alborzi (2008)* also, search for nearby points of interest using road network distance using keyword query. However, a limitation is the lack of support for other valuable metrics, such as travel time or temporal considerations. In recent years, there has been a growing emphasis on developing spatiotemporal databases capable of handling massive datasets with diverse temporal characteristics. Temporal queries retrieve

query results based on a specified temporal or time setting with spatial data. Notably, the temporal dimension exerts a significant influence in various domains. Numerous works have been studied and proposed in this context, including (*Fan et al., 2010*; *Yuan et al., 2013*).

*Fan et al. (2010)* introduced a type of solution for incorporating a time dimension, while *Yuan et al. (2013)* proposed a method for providing time-aware recommendations using snapshots and events approach. The integration of temporal aspects into spatial databases has become increasingly critical as it enables the representation and analysis of data that evolve over time. This intersection of temporal and spatial data has broad applications, including tracking the movements of objects in space over time, monitoring environmental changes, managing transportation systems, and examining the interconnected of people and places in large metropolitan cities (*Hoang-Vu, Vo & Freire, 2016*). Furthermore, advances in sensor technologies have led to the generation of extensive spatiotemporal sensor data, making efficient data management essential. The work of *Breunig et al. (2020)* focuses on the integration of temporal data from IoT sensors into spatial databases, contributing to improved decision-making in applications like environmental monitoring. The efficient management of temporal queries within spatial databases is of paramount importance, not only for researchers in the field of geographic information systems (GIS) but also for professionals seeking precise analysis of temporal-spatial data in various applications (*Worboys & Duckham, 2004*). Efficient indexing accelerates query processing within trajectory and temporal-spatial databases (*Deng et al., 2011*). One widely adopted indexing technique is the quadtree (*De Berg, 2000*), a hierarchical spatial index that partitions space into quadrants. Quadtree-based indexing is particularly well-suited for spatiotemporal data due to its capacity to efficiently manage both spatial and temporal dimensions (*Chen, Cong & Cao, 2013a*). The concept of quadtree indexing was first introduced by Raphael Finkel and J. L. Bentley in 1974 (*Waresiak & Skrzyński, 2011*). In a quadtree index, each node represents a spatial region at a specific temporal interval (*Eldawy et al., 2015*). It is employed to store two-dimensional spatial data in a tree structure. This two-dimensional space is recursively subdivided into four quadrants, as illustrated in Fig. 1. Each tree node has either zero or four children, and spatial information is stored in leaf nodes. This hierarchical structure facilitates rapid data retrieval within a specified spatiotemporal range. By recursively subdividing spatial regions based on occupancy and time, quadtree indexes support not only range queries but also more complex spatiotemporal queries, such as nearest-neighbor searches and trajectory-based queries. Researchers have extended the basic quadtree concept to create variants optimized for specific types of spatiotemporal queries, thereby enhancing the versatility and performance of this indexing approach (*Finkel & Bentley, 1974*; *Mahmood, Punni & Aref, 2019*). The utilization of quadtree-based indexing has thus become a cornerstone in the effective management and retrieval of temporal-spatial data, enabling advanced query capabilities across a wide range of applications (*Mokbel, Ghanem & Aref, 2003*).

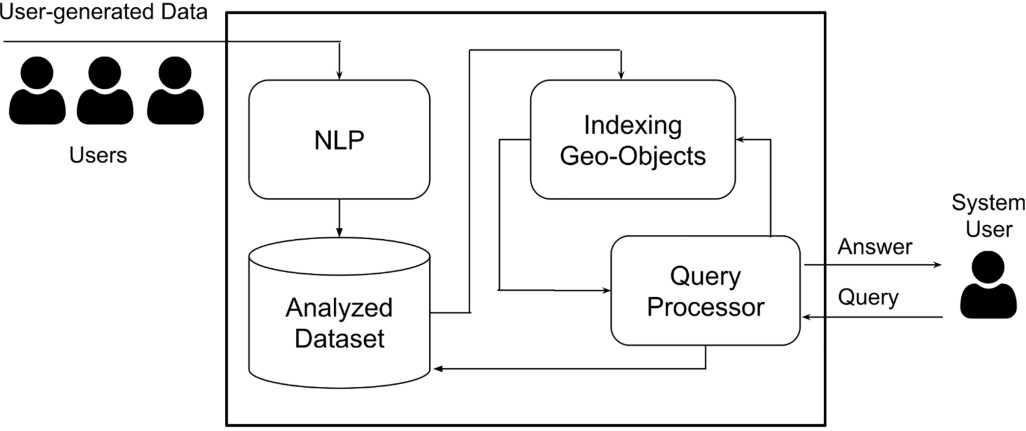

**Figure 1  The geo-sentiment analysis framework.**

## PROBLEM STATEMENT

Sentiment analysis is a powerful approach to understand people's opinions and thoughts that is incomplete without considering location and time. For example, social media sentiment analysis becomes more useful for businesses when focusing on their surrounding neighborhoods and the latest posts. Our work enhances content sentiment analysis by including spatial and temporal aspects. It enables analyzing opinions specifying time and location, such as the latest 100 posts in Riyadh City regarding specific topics.

To achieve that objective, we identify our research problem as follows. Sentiment analysis queries, *GeoSentiment*, are evaluated on a geo-textual dataset $D$ that consists of a set of geo-textual objects. Each object $o \in D$ is represented with (*loc, kw, time, sentiment*), where *loc* is a point location (latitude/longitude coordinates), *kw* is a set of keywords, *time* is a timestamp, and *sentiment* is the sentiment score of the object base on *kw*. $D_{t_1}$ is a snapshot of the dataset $D$ at time $t_1$, so every object $o \in D_{t_1}$ has $o.time \leq t_1$. Table 1 gives an example of a dataset that consists of eight objects, $o1$ to $o8$, each is associated with a set of keywords, a timestamp, and sentiment score which could be range from −1 to 1, where negative scores indicate a negative sentiment while positive scores indicate a positive sentiment.

To attain this goal, we define our research problem as follows: Sentiment analysis queries, denoted by *GeoSentiment*, are assessed using a geo-textual dataset $D$ comprising a collection of geo-textual entities. Each entity $o \in D$ is described by (*loc, kw, time, sentiment*), where *loc* represents a point location (latitude/longitude coordinates), *kw* denotes a set of keywords, *time* signifies a timestamp, and *sentiment* denotes the sentiment score of the entity based on *kw*. $D_{t_1}$ represents a snapshot of the dataset $D$ at time $t_1$, implying that every entity $o \in D_{t_1}$ has $o.time \leq t_1$. Table 1 illustrates an instance of a dataset containing eight entities, $o1$ to $o8$, each associated with a set of keywords, a timestamp, and a sentiment score ranging from −1 to 1, where negative scores indicate negative sentiment and positive scores indicate positive sentiment.

Given a *GeoSentiment* query $q = (w, r, k, t)$, where $q.w$ is a set of keywords, $q.r$ is a spatial region, $q.k$ is an integer, and $q.t$ is a timestamp, $q$ finds $k$ objects $o_i \in D_t$, $1 \leq i \leq k$, such that: (1) $o_i.kw \cap q.w \neq \phi$, (2) $o_i.loc \in q.r$, and (3) $o_i$s are the most recent $k$ objects in

**Table 1 Sample of objects in the dataset.**

| ID | Location | Keywords | Timestamp | Sentiment |
|----|----------|----------|-----------|-----------|
| $o1$ | −77.03, 38.89 | Final, Cup, Ceremony, Fun | 01-02-2024 20:18:30 | 0.95 |
| $o2$ | −60.53, 30.70 | Inspiring, Openning, Speech | 01-02-2024 20:18:26 | 0.8 |
| $o3$ | −78.55, 40.89 | NBA, Lakers, Loss | 01-02-2024 20:18:20 | −0.5 |
| $o4$ | −63.73, 29.90 | World, Open, Tennis, R.Nadal, D.Thiem | 01-02-2024 20:18:19 | 0.1 |
| $o5$ | −50.88, 20.89 | Awful, Pizza, Taste | 01-02-2024 20:18:15 | −0.8 |
| $o6$ | −10.03, 29.08 | Stock, Market, Bull | 01-02-2024 20:18:10 | 0.9 |
| $o7$ | −40.66, 41.89 | Brazil, FIFA, Argentina, Game | 01-02-2024 20:18:05 | 0.2 |
| $o8$ | −51.77, 24.60 | NBA, LeBron, Injury | 01-02-2024 20:18:00 | −0.6 |

$D_t$. So, $q$ retrieves $k$ objects from the dataset snapshot $D_t$ that corresponds to the query timestamp $t$. Then, the average sentiment score is calculated to evaluate the sentimental of the given query predicates. Each object lies in the spatial distance query and contains one or more of the query keywords. In addition, the $k$ objects are ranked based on time to retrieve the most recent objects in $D_t$. This article aims to use proper indexing techniques to answer this query type efficiently to provide spatial-temporal sentiment analysis.

The overall framework is shown in Fig. 1. Basically, it consists of four different main components. NLP is the module that analyzes the user-generated data to determine whether the given object is positive, negative, or natural. The literature has number of NLP techniques that can be adopted (*Qiu et al., 2020*; *Medhat, Hassan & Korashy, 2014*). The output of the NLP processing is fed to the central data warehouse where the data is ready to be queried. When the user submits a query, a simple query is triggered to fetch the relevant data from the data warehouse with respect to interval time (*Purves et al., 2007*). The fetched data is indexed by the geo-index component in batched fashion. Finally, the query processor utilizes the geo-index to efficiently return the sentiment analysis result with respect to the submitted user query predicates. The main contribution of this work is the indexing geo-object and the query processor components. The following sections detail these components.

## INDEX STRUCTURE

Our solution proposes two types of indices to process spatial distance queries providing spatial-temporal sentiment analysis. The first is a basic index linking posts with their locations, and the second is a hybrid index supporting keyword searches.

### Basic spatial indexing

We use a simple spatial index, namely Quadtree (*De Berg, 1997*; *Samet, 1984*), to link posts with their locations, supporting geospatial queries. We index each post and its sentiment score according to its location as a data point in a Quadtree data structure. As a result, spatial queries can be processed efficiently using this index.

In a Quadtree, each node may have zero or four child nodes, hence its name. This structure works well for spatial requirements as it divides a two-dimensional space

recursively into four equal quadrants. Additionally, a Quadtree has a bucket capacity, determining the maximum number of data points that can be stored in a single node. Consequently, setting a large bucket capacity value reduces the depth of the tree and vice versa (*De Berg, 2000*).

Like B+ trees (*Bayer & McCreight, 1970*; *Knuth, 1973*), data points are stored in the leaf nodes only, while internal nodes serve as pointers. An insert operation navigates the tree until it reaches the proper leaf node. The data point is added if the leaf's node bucket capacity is not reached. Otherwise, the leaf node is split into four children, and the data point is added to the proper node. A delete operation works similarly to find a data point and remove it. Figure 2 is shown the general structure of Quadtree where the object reside on the leaf nodes. The time complexity of tree operations depends on the tree's maximum depth. Insert, delete, and search operations have logarithmic time complexity, with potentially linear time for extremely unbalanced trees (*De Berg, 1997*).

The process of indexing a large number of posts may take a considerable amount of time. We address this issue by inserting posts in batches instead of single inserts. We construct Minimum Bounding Rectangle (MBRs) based on incoming posts to group nearby posts and perform batched inserts. Each post group is inserted as one batch to its corresponding leaf node.

The MBRs are dynamically created and periodically updated based on the location of incoming posts. This process is cascaded until the leaf nodes to group posts further. The MBRs of high-level nodes are larger than lower nodes as areas become fine-grained, going deeper in the tree. Specifically, each node, except leaf nodes, has a dynamic MBR to combine all incoming posts within the boundary of its child nodes.

### Hybrid index

This hybrid index extends the basic spatial index to provide keyword-based searches. It contains a Quadtree, similar to the basic index. However, each leaf node of the Quadtree references an inverted index containing a hashtable. The hashtable consists of key-value pairs mapping keywords to a list of posts and their sentiment scores. This list is sorted in reverse chronological order from newest to oldest to support top-$k$ retrievals. The added layer of inverted indices facilitates keyword lookup for spatial queries. Figure 3 is shown the hybrid index where the leaf nodes store an inverted index as additional layer compare to the basic index.

## QUERY PROCESSING

This section details the query processing of *GeoSentiment* defined in the "Problem Statement" section utilizing the proposed basic and hybrid indexes introduced in the "Index Structure" section. In general, the query processing retrieves the top-k objects from the spatial index based on the structure of the index while employing the pruning techniques based on the underlying index structure. The sentiment analysis scores are embedded in each object. Therefore, the sentiment scores do not play an essential role in any pruning techniques compared to the spatial, temporal, and keyword attributes. The following subsections detail the query process for each index, basic and hybrid indexes.

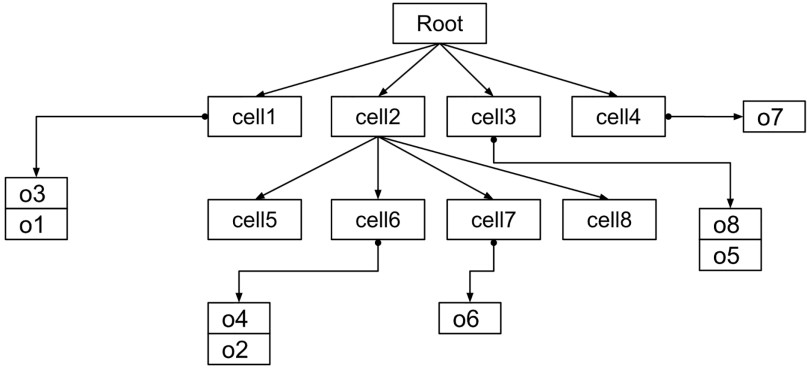

**Figure 2** The basic index structure.               

## GeoSentiment query using the basic index

The query processor starts by the spatial predicate value which represents the MBR region. This MBR is used to locate the objects that their spatial value overlaps with the query MBR. The objects are organized in the index structure using Quadtree which distributes the objects into the leaf nodes of the Quadtree based on the objects spatial values. The leaf node contains a list of objects ordered by the timestamps. The last object has the most recent timestamp while the first object in this list has the oldest timestamp within this node.

The query processor performs the following steps in order to get top-$k$ objects that match the query predicates:

- **Step 1:** The query processor starts from the root node and navigates the Quadtree into the internal levels until reaching the leaf nodes. All leaf nodes that overlap with MBR of the query will be inserted into a priority queue data structure ($Q$) based on the timestamp of the last (the most recent) object in the list.

- **Step 2:** An initial query result ($QR$) is constructed using a hashtable data structure. The list which has the object that has the most recent timestamp in the priority queue ($Q$) is dequeued. Then, the leading object is removed and inserted in the $QR$ if the object contains one of the query keywords and the list (if it is not empty) is enqueued back to the priority queue ($Q$). This step is repeated until $K$ objects that satisfy the query predicates retrieved or the $Q$ is empty which means all objects have been retrieved and checked against the query predicates but less than $K$ objects satisfied the query objects. It is worth noting that the structure of the basic index does not support any keyword indexing structure. Thus, the full scan of all objects is the only option to filter out the objects that match the query keywords predicates.

- **Step 3:** To calculate the sentiment analysis of the objects in $QR$, we simply retrieve the objects one by one and sum the sentiment sores. Then, the average is calculated based on the sum of the sentiment scores and the length of the $QR$.

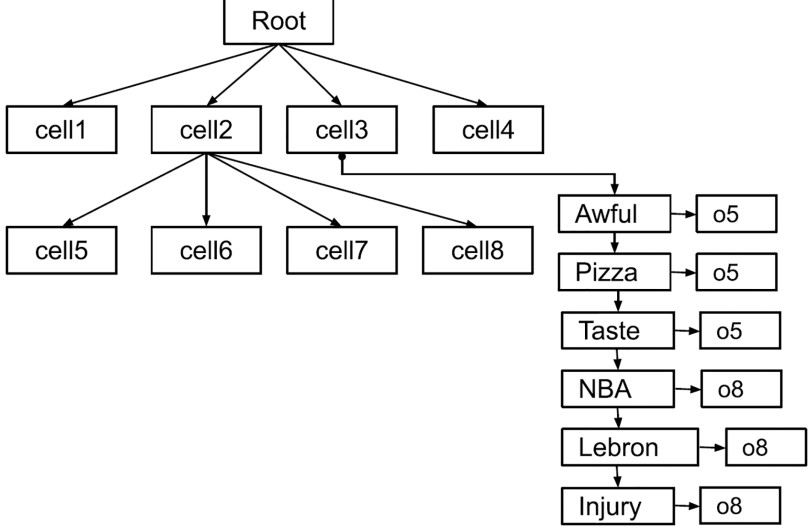

**Figure 3 The hybrid index structure.**

## GeoSentiment query using the hybrid index

The query processor using the hybrid index performs the same steps as using the basic index except in Step 1. Since the objects in the leaf nodes are organized by the inverted index, the query processor elevates only the lists that contain the query keywords by retrieving these lists utilizing the inverted index. Thus, retrieving $K$ objects is significantly faster than the basic index.

## EXPERIMENTAL EVALUATION

An experimental evaluation of the aforementioned query processing methods and indexes is provided in this section. The evaluation includes memory consumption, data ingestion, and query evaluation with varying settings. Specifically, we used the following metrics to measure system performance: 1) memory usage in GB, 2) indexing latency (*i.e.*, the time needed to construct the index), and 3) query latency (*i.e.*, the time needed to retrieve query results).

### Experimental setup

We evaluated indexes and their query processing and using an Intel(R) Core (TM) i7-8550U CPU @ 1.80 GHz 1.99 GHz and 8 GB RAM running Windows 10 (64 bit). The evaluation datasets and query workloads are described below.

The parameters are defined as follows: dataset size, query answer size ($k$), query distance ($R$), and number of keywords. Default values are assigned to each parameter; the default dataset size is 5 million objects, the default query answer size ($k$) is 100 objects, the default query distance ($R$) is 50 km, and the default number of keywords is two. All experiments utilize Java 8 implementations for the evaluated indexes and their query processing, and run on an Intel(R) Core (TM) i7-8550U CPU @ 1.80 GHz 1.99 GHz with 8 GB RAM, operating on Windows 10 (64 bit). Details regarding the evaluation datasets and query workloads follow.

## Datasets

The dataset has been collected from Twitter platform by using Twitter API as compressed JSON files. Around 20 million tweets have been collected over the course of 5 days. These tweets were pr-processed to become ready for working on it. A script has been written to parse the JSON files and to pre- processing this dataset. These tweets will be filtered according to the pre-computed location (determined using data points' coordinates) and language. Then, the text of tweets was tokenized by replacing spaces and commas between words by commas ",". During the data collection process, tweets are inserted in batches and MBRs are generated based on incoming tweets in order to group nearby tweets. As a result, each tweet is assigned a MBR region. Also, the centroid is calculated for each tweet from its MBR to latitude/longitude coordinate values to make the checking easier if the given tweet belongs to the bounding box of specified location. After that, the sentiment analysis was used to calculate the sentiment score for each tweet. This is done by using Stanford NLP Library of Sentiment Analysis (*Socher et al., 2013*). At the end, the extracted tweets were stored in a stoarge such as a data warehouse where each tweet consists of id, latitude/longitude coordinate of the tweet, NLP score, and the tokenized text of the tweet. A substantial portion of the collected tweets is eliminated due to the lack of location information and the use of non-English language. The extracted number of tweets after pre-processing is around 5 million tweets.

## Query workloads

The query workload has been generated to create multiple queries for testing the indexes and a range query for these indexes. Different 1,000 coordinates of different points within the specific location are sampled from real location queries of Bing Mobile users (*Magdy et al., 2014b*). For each query, six different keywords are taken randomly from the text of the objects. Each query in the output consists of latitude/longitude coordinates that represent the location and six different keywords. The generated queries are stored in a text file to be used in testing and evaluation of the indexes and the query processing techniques.

### *Memory consumption*

Figure 4A displays the memory usage of two types of indexes: the basic spatial index and the hybrid index consisting of a spatial index and a keyword inverted index. The dataset sizes were varied during the analysis. Modifying the size of the dataset has an impact on the allocation of memory resources. Figure 4B demonstrates a linear increase in memory resources for both types of indexes. When the dataset consists of 2 million objects, the basic index consumes 0.5 GB. If the dataset size doubles, the same index consumes 1.01 GB. Analogously, the hybrid index exhibits the same behaviour. Consequently, the hybrid index consistently requires a larger amount of memory compared to the basic index.

### *Data ingestion*

This section evaluates the indexing speed of both basic and hybrid indexes in relation to different dataset sizes. Figure 4B demonstrates that the speed of indexing increased in a linear manner for both types of indexes. When the dataset size is 2 million objects, the basic index requires 8.4 msec to index the objects, whereas the hybrid index takes

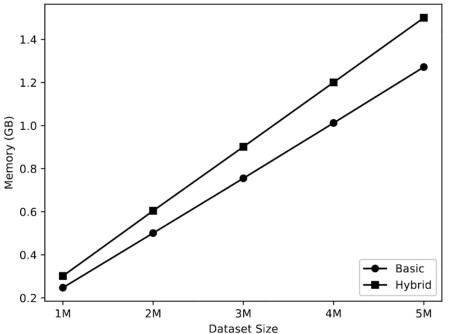

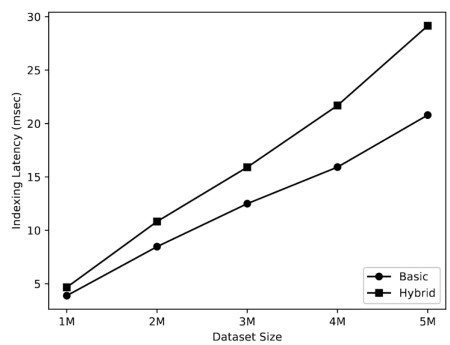

(a) Memory Consumption with Varying Dataset Sizes.     (b) Indexing Latency with Varying Dataset Sizes.

**Figure 4** Query workloads.   

10.8 msec for the indexing process. As the dataset size grew to 4 million objects, the time required for indexing also increased. Specifically, the basic index took 15.9 msec, while the hybrid index took 21.69 msec to index the objects. The overall outcome indicates that the hybrid index consistently requires more time for indexing compared to the basic index. Specifically, we noticed a a linear increase in indexing latency (*i.e.*, time required to construct the index) for both types of indexes.

## Geo-sentiment query evaluation

This section presents the assessment of *GeoSentiment* Query in relation to range queries. The evaluation focuses on the utilization of both basic index and hybrid index to search for keywords within these indexes. This evaluation assesses the querying process in each index, taking into account the different values of the query result $k$, the range of the query $R$, and the number of keywords to be searched.

Figure 5A illustrates the impact of different values of $k$ on the query time of *GeoSentiment* Query. According to the figure, the query time rises as the value of $k$ increases due to the need for additional processing to obtain a larger answer. Nevertheless, the query time in the basic index is greater than the query time in the hybrid index. The hybrid index demonstrates superior performance with query time of 70 milliseconds (msec) when $k = 10$. However, this query time increases to 445 msec when the value of $k$ is changed to 1,000. On the other hand, the initial query time of the basic index is 397 msec when $k = 10$, and it increases to 9,310 msec when $k = 1,000$. Therefore, we can conclude that the query time in a hybrid index is significantly faster, exceeding the query time in the basic index by more than 20 times.

### *Effect of varying ranges R on geo-sentiment query*

Figure 5B illustrates the impact of different range values $R$ on the query time of *GeoSentiment* Query. As depicted in the diagram, the query time increases as the range value increases due to the additional processing required to obtain a larger response. Nevertheless, the query time in the basic index is greater than the query time in the hybrid index. According to the figure, the hybrid index demonstrates the highest performance, with a query time of 170 milliseconds (msec) at a distance of 10 km. However, this query

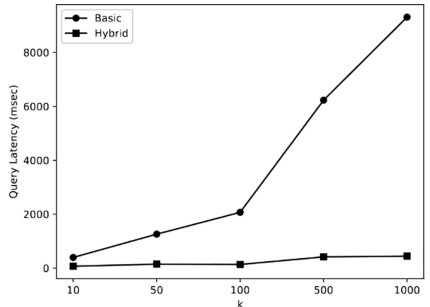

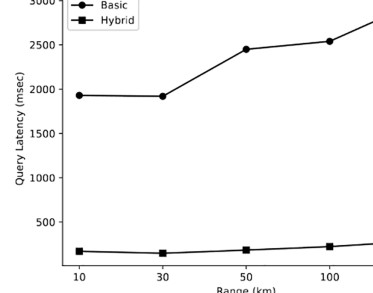

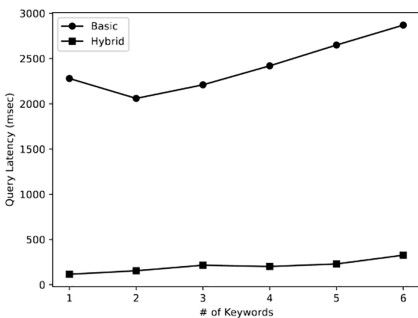

**(a)** Geo-Sentiment Query Time with Varying k.

**(b)** Geo-Sentiment Query Time with Varying Ranges.

**(c)** Geo-Sentiment Query Time with Varying Keyword Numbers.

**Figure 5  Geo-sentiment query evaluation.**               

time increases to 282 msec when the range value is changed to 200 km. On the other hand, the initial query time in the basic index is 1,930 msec when the range is 10 km, and it increases to 2,960 msec at a range of 200 km. Therefore, we can deduce that the query time in a hybrid index is nine times faster than the query time in a basic index.

### Effect of varying keyword numbers on geo-sentiment query

Figure 5C demonstrates that the query time exhibited an increase as the quantity of keywords to be searched grew. For instance, the primary index requires 2,280 msec to search for a single keyword, and this duration increases to 2,870 msec when searching for six different keywords. The query time in a hybrid index is 116 msec when searching for a single keyword, and this value increases to 327 msec when searching for six keywords. The query time in the basic index is nine times greater than the query time in the hybrid index.

## DISCUSSION

In this section, we discuss the methodological choices we adopt in the design and implementation of our GeoSentiment query system. We also provide discussion on the key results and findings.

### Keyword inclusion *versus* exclusion

As outlined in our problem statement, the *GeoSentiment* query must include keywords, adhering to the formulation $o.kw \cap q.w \neq \varnothing$. Here, $o.kw$ denotes the keyword set tied to an object $o$, while $q.w$ represents the keywords within the query, ensuring this intersection is non-empty. This approach aligns with the standard practices observed in the related work leveraging sentiment analysis on spatio-temporal data (*Hu et al., 2019*). We adopt this choice to ensure that the results of the query encompass objects that are not only relevant to the query but also carry meaningful insights from social media data, especially in such analytical problems with geographically specific contexts.

## Exclusion of kNN queries

Our decision to exclude $k$-nearest neighbor (kNN) queries from our study was primarily justified by the fact that $k$NN queries do not align well with the nature of distance searches, which is the core of our study. In distance query, the goal is typically to retrieve objects within a defined spatial boundary; while in $k$NN query, the goal is to find the $k$ closest objects to a specified spatial point. $k$NN queries offer limited value since sentiment analysis is often used to capture the aggregate emotional tone within a specific region and context. Moreover, combining $k$NN and distance queries would introduce unnecessary complexity and require additional processing that is not essential for achieving our objective of capturing all relevant data within the specified distance (*Almaslukh & Magdy, 2018*).

## AND operator *versus* OR operator

Our approach features the use of the OR operator in the keyword matching queries, as opposed to the AND operator. This decision is captured in the formulation of our problem definition, *i.e.*, $o.kw \cap q.w \neq \varnothing$. The use of the OR operator allows for a broader retrieval of data, ensuring that any tweet containing at least one of the specified keywords is considered for analyzing its sentiment. In contrast, the use of the AND operator would limit the tweets retrieved to the ones that contain all the keywords in the query. Given short text of tweets and often sparse nature of social media data, such an approach could lead to a substantial reduction in the data retrieved, thereby limiting the comprehensiveness and utility of our analysis. Moreover, the computational cost associated with the AND operator is considerably higher, as it requires more complex query processing and repeated index scanning (*Cary, Wolfson & Rishe, 2010*).

While our work focuses on the OR operator to maximize data retrieval and ensure a comprehensive analysis, we acknowledge the potential benefits of incorporating both OR and AND operators in future work to balance comprehensiveness and specificity, thereby enhancing the overall robustness of our approach.

## Application of different NLP techniques

In our study, the natural language processing (NLP) component, as depicted in Fig. 1, plays a critical role in computing the sentiment scores of tweets. However, it is important to acknowledge that processing large volumes of tweets through this NLP component may add significant computational overhead to the system. This is primarily attributed to the computational resources required to parse, understand, and compute the sentiments expressed in natural language. We make it clear that optimizing the NLP component is not within the scope of our study. Our focus is primarily on the application and effectiveness of sentiment analysis over spatio-temporal social media data.

## Limitations and future work

Based on the presented results, it is clear that the hybrid index takes more space in memory to indexing objects, this is due to the use of inverted indexes to index the tweets according to the keywords they include. Additionally, the result shows that the hybrid index takes more time to index objects compared to the basic index as the inverted index in each leaf

node has to be built. In terms of query processing performance, employing the hybrid index contributed significantly to reducing query time to less than 10% the query time of experienced when the basic index is used. This reduction is observed when varying different parameters, including, $k$ query result numbers, range of the query, and number of keywords to be searched.

In future research, the Lexicon Based Approach of sentiment analysis could be employed to decrease the computational time required for determining the sentiment score of the objects in the dataset. In order to enable the consolidation of processing data code with the indexing and querying model, a unified model could be created to directly handle the streaming data. The quadtree index can also be utilized to retrieve the top-k objects that are closest to a specific point through the KNN query.

## CONCLUSIONS

This article presents GeoSentiment, a novel analytical query for effectively performing sentiment analysis on user-generated data, while also considering spatial and temporal aspects. This query can assist enterprises in various problem settings by facilitating data processing, enabling faster response to potential risks, and facilitating the creation of more robust analytical queries. This study employs the range query to compute the sentiment analysis score for the top-$K$ geographical objects that satisfy the keyword, time, and region conditions. The experimental results indicate that the hybrid index structure surpasses the baseline index, which solely indexes spatial aspects without considering object keywords. The hybrid index significantly decreases query latency by an order of magnitude. The enhancement was achieved through the utilization of a hybrid index structure and the implementation of pruning techniques in the query process.

### Funding
This work was supported by the Researchers Supporting Project number (RSP2024R451), King Saud University, Riyadh, Saudi Arabia. The funders had no role in study design, data collection and analysis, decision to publish, or preparation of the manuscript.

### Grant Disclosures
The following grant information was disclosed by the authors:
King Saud University, Researchers Supporting Project: RSP2024R451.

### Competing Interests
The authors declare that they have no competing interests.

### Author Contributions

- Abdulaziz Almaslukh conceived and designed the experiments, performed the experiments, analyzed the data, prepared figures and/or tables, and approved the final draft.

- Aisha Almaalwy conceived and designed the experiments, performed the experiments, performed the computation work, authored or reviewed drafts of the article, and approved the final draft.
- Nasser Allheeib conceived and designed the experiments, authored or reviewed drafts of the article, and approved the final draft.
- Abdulaziz Alajaji performed the computation work, prepared figures and/or tables, authored or reviewed drafts of the article, and approved the final draft.
- Mohammed Almukaynizi performed the computation work, prepared figures and/or tables, authored or reviewed drafts of the article, and approved the final draft.
- Yazeed Alabdulkarim conceived and designed the experiments, prepared figures and/or tables, authored or reviewed drafts of the article, and approved the final draft.

### Data Availability

The data and code are available at GitHub and Zenodo:

- https://github.com/aalma021/SentiAnalysis.git

- Almaslukh, A. (2024). Top-k sentiment analysis over spatio-temporal data. Zenodo. https://doi.org/10.5281/zenodo.13335363.

The dataset was collected *via* Twitter API URL: https://developer.x.com/en/docs/twitter-api.

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
