# Peer review of "Top-k sentiment analysis over spatio-temporal data"

_PeerJ Computer Science, doi:10.7717/peerj-cs.2297_

## Round 0.1 · original submission · Minor Revisions

Dear Authors, as you could see given the time that passed by, it was difficult to find reviewers over the past months. However, we finally found 2 reviewers that agreed and I did a further in-depth review.

Considering Reviewer 1's comments, I agree on that it may be good to add a little more literature. My personal concern here is that you do not outline current (Spatial) DB capabilities (e.g. Postgis, Oracle, SQL Server... for a list see Steiniger and Hunter, 2012 overview on FOS GIS and or the SDI overview), Also, you don't describe the Twitter API used (i.e. the parameters it has). Perhaps it would be also good to know what search options the Instagram API has. Because, honestly, I thought these Social media provide APIs that allow spatio-temporal queries.

Regarding Reviewer 2, I also agree with him that it is of utmost importance to have instructions and reproducible code provided. So, please check and add instructions to the github repository. Ideally, if someone else can test with your new instructions.

Finally, my own comments (see also the attached pdf):

General Questions/Suggestions:
========================
- revise the wording: instead of “range query", use “(spatial) distance query”
- instead of "latency", use query time. Otherwise explain why it is better to use latency here.
- revise Section numbers/names and in-text citation style (it seems like copied from another style)
- In the beginning it was not clear to me that you would propose two solutions and then compare them, so please make this more clear at the outset (introduction, etc.).
- The distance query in the tests is given in km, so in what coordinate system are the geometries stored? I would expect that Twitter data comes with geographic coordinates (for which we can't calculate distances in km, but only degrees). Please explain if you do/did conversions or so.
- I miss a survey on spatial databases and if they provide spatio-temporal indexing, such as mentioned here: https://stackoverflow.com/questions/67905589/how-to-query-space-time-points-with-index-in-postgis and/or here: https://blog.rustprooflabs.com/2020/11/postgis-trajectory-intro
- Also two books may need to be added with relevant content on Quadtrees and on Temporal queries: one is the spatial geometry book by de Berg, Cheong, van Kreveld & Overmars (2008), who have an entire section on quadtrees, and the other is Worboys & Duckham (2004) - which also addresses time queries.
- I miss some information about future work, e.g. implementing AND operations, or a combined K-NN query with distances, etc.
- To safe space consider to group the results graphics from the experiments (e.g. combine 2 into 1, or 4 into 1).

Detailed Comments:
===============
Introduction:
- l.28: please add a reference to a book or so that includes some of these examples.
- l.29: add examples for huge datasets at a rapid rate, i.e. Twitter, Instagram etc.. It is not clear to me of what other data one could think of in an enterprise context.
- l.31: “Sentiment analysis is considered as one of the main building blocks “ => considered by whom? (reference)
- l.38: “and react to any potential reputation risks” => well this sounds strange to talk of “reputation risk”. I think it’s rather important to react. Seeing this from a perspective of e.g. a transport network incident (e.g. a metro line that does’t run, or so). Another application field is tourist attractions, but this usually in a positive way, so there is no reputation risk.
- l.48: “the result of the queries is normally huge in size“ => what means “huge”? Please give examples. I actually think the result set may be small if its over 3 predicates
- l.50: 10.000? How many responses did you get for this query, and in what platform?
- l.59: Latency: Please give an example for a need to do real time analysis.
- l.70: Here I stumbled the first time over the term range query and asked myself/And you: What is a “range query”? => later on I figured that you mean a spatial distance query. So I was thinking of a query type first, with a name used in computer science, that I as an engineer may not have heard of before.
- l.78: What means Scalable? Not sure what is new about the first point? Didn’t the Twitter API do this already? Based on your writing /experiment later on I guess Twitter doesn’t?
- l.80 “smartly prune” => Why “smart”? (or why dumb?)
- Please edit this section on the citation style… its all messed up with/without the brackets in the wrong positions. Similar holds for the section numbers, that seem to have been used in an earlier version, while there are no section numbers here.

Related Work:
- l.90: which query type did Cao et al. introduce?
- l.95 on Samet et al. (2008): really, he did not foresee this? (I thought he did publish on this too, but maybe I am wrong)
- l.123 Figure 1 does not show the Quadtree but figure 2 does.
- l.128 “Researchers have extended the basic quadtree concept to create variants optimized for specific types of spatiotemporal queries,“ => please add at least 2 references
- Also, When you describe the framework or in this section I wonder if you can add a reference to Purves et al. (2007, IJGIS) as one of the first spatial and text combining search engines. I am not one of the authors though. He may have also a further relevant article on Flickr (Hollenstein and Purves, 2010). Also when talking about the quudatree further below you may add a reference to the well-known spatial geometry book by de Berg, Cheong, van Kreveld & Overmars (2008), which has full chapter on Quadtrees. Even more so, the book by Worboys and Duckham (2004) talks in chapter 10n about “Time” and Section 10.4 is about time indexing queries. So you may need to cite/add this.
- Purves, R. S., Clough, P., Jones, C. B., Arampatzis, A., Bucher, B., Finch, D., ... & Yang, B. (2007). The design and implementation of SPIRIT: a spatially aware search engine for information retrieval on the Internet. International Journal of Geographical Information Science, 21(7), 717-745.
- Another thing is, that I wonder why you didn’t consider the R-Tree instead of a quadtree.
- Please add a statement if one of the spatial databases such as Oracle, Postgis, SQL server, or big data DBs such as GeoMesa (or Lucene?) provide perhaps already spatial - temporal indexing and query methods? If this would be the case it may be tricky though to justify your work. However, if this is not the case, then even better. Note that I actually just found this: https://stackoverflow.com/questions/67905589/how-to-query-space-time-points-with-index-in-postgis And this: https://blog.rustprooflabs.com/2020/11/postgis-trajectory-intro

Problem Statement:
- l.153: Order isn’t clear: first the NLP, then time query, and then location query? Is the spatial indexing done after retrieving the time query results? Please clarify.

Index Structure:
- l.167: “Our solution offers…” => I am not sure here if you propose 2 solutions, each one using a different
index, or if both indexes are part of the same solution. Please clarify.
- l.170: “Basic Index” => perhaps better “Basic Spatial Indexing”?
- l.171: Add also reference to the Spatial Geometry book by de Berg et al. for the quad179trees
- l.179: you refer to a B+ tree, but the reader was not introduced to it. So add a Reference at least, or remove, etc.
- l.184: “search operations have logarithmic time complexity, with potentially linear time for extremely unbalanced trees” => please add a reference to this statement (maybe the de Berg et al. book can help?)
- l.187: “We construct Minimum Bounding Rectangle (MBRs) based on incoming posts” => please see my comment further below on MBR construction on tweets/messages
- l.196 “This index…” => please specify what index.

Query Processing:
- l.215: “The leaf node contains a list of objects ordered by the timestamps.” => so to clarify: Each leaf has a list of objects ordered by Time?
- l.224: “QR”: What stands QR for?
- l.225: “queue Q its list is” => “queue Q in the list is”?
- l.236 Section title “GeoSentiment query” => So, this is a second solution to the same problem? Hence, you propose 2 different solutions to compare later? (…looks like it, when I keep reading) => If this is the case, this needs to be mentioned more clearly in the introduction. I didn’t get this on my first read.

Experimental Evaluation
- l.249: You write that you implemented this in Java. Though, I wonder if some existing code was used (to give their authors credit), e.g. JTS / GEOS code is the code available?
- l.257: Please explain where the location information came from? Was this obtained from NLP or from the phone GPS coords?
- l.258: How did you calculate a centroid and MBR from a tweet, if it is a point information? - or what is meant with “specified location” - e.g. a city?
- l.264: Please add what happened to the other 15 Million tweets? Was location information missing for those?
- l.271: makes reference to Figure 6.1 => There is no Figure 6.1
- l.280: you use the word “phenomenon” but generally the term is used for not explainable things, so I suggest you use “behaviour”.
- l.289: ok, But the more important question is, if the indexing speed varies (more or less) linear with dataset size? Please add a statement in this.
- l.292: You never explained what a range query is. So how should the reader understand? - Further below I see this query is in km, so I think what you mean is a spatial (distance) query. Please change this.
- l.297: k is the results set size? Please add, as this is stated nowhere in this section.
- I think, to save space, it may be good to group some figures. E.g. have figure 4 and 5 in one figure X, left and right.
- Maybe, as there isn’t so much information in the graphics, you can group 2 or 4 figures into one (but two may be better)?
- l.304: “latency….is significantly faster, exceeding the latency of the query” => “query time is significantly faster, reducing the query time with the basic index by a factor of 20“; latency cannot be faster, it can only be shorter or longer

Discussion:
- l.333: Exclusion of kNN Queries => ok, thats an interesting perspective. However, I would have thought that you would combine them and implement whatever threshold (distance or k) comes first.
- l.340: AND Operator versus OR Operator => I think, ideally, one should enable both in the future: OR and AND queries. For instance in a transportation related event, some may write “metro” while other write a different common household name. So one needs to combine event tweets.

Conclusions:
- How about future work? What should/needs to be addressed?

Reviewer 1 ·

Basic reporting

1) It is suggested to write the problem statement to the point and support your problem statement from the past research.
2) Increase the Literature Review, Add the Table containing, Sentiment old research approach, result and their limiations.

Experimental design

1) It is suggested to add the section of performance measures which you used in your research.
2) The authors used the real time data of "X". it is suggested to use the some benchmark data of twitter and X and run their model of that dataset along with real time dataset.

Validity of the findings

1) Add confusion matrix, ROC curvues to support the results.
2) Choose some baseline appraches and perform the comparsion and discuss, why your approach / model is better from baselines.

Cite this review as
Anonymous Reviewer (2024) Peer Review #1 of "Top-k sentiment analysis over spatio-temporal data (v0.1)". PeerJ Computer Science

Reviewer 2 ·

Basic reporting

No comment

Experimental design

No comment

Validity of the findings

I was unable to replicate the findings as there was no guidance provided on how to run the software on the Github provided

Cite this review as
Anonymous Reviewer (2024) Peer Review #2 of "Top-k sentiment analysis over spatio-temporal data (v0.1)". PeerJ Computer Science

---

## Round 0.2 · accepted · Accept

Thank you for this revision. The authors have addressed my comments to my satisfaction. The manuscript (on a quick glance) is more or less ready to be published. I found two issues:

- l.179: Reference citation should be (Purves et al. 2007) => also requires fix in Bibliography regarding names, surnames, etc.
- l.264: two sentences are weirdly merged in the revision.
(perhaps another read (by a native english speaker) could do well.)

Also reviewer 2 mentioned that he had some problems with the code provided. So please check if some improvements are needed here. (Reviewer 2 comment has been: "I was asked just to check code functionality. The Readme added by the authors points to a directory that, unless I'm mistaken, doesn't exist ("Final Project/..."). As such, I was unable to run the code")

Thanks,
Stefan Steiniger

Reviewer 1 ·

Basic reporting

no comment

Experimental design

no comment

Validity of the findings

no comment

Additional comments

no comment

Cite this review as
Anonymous Reviewer (2024) Peer Review #1 of "Top-k sentiment analysis over spatio-temporal data (v0.2)". PeerJ Computer Science

Reviewer 2 ·

Basic reporting

-

Experimental design

-

Validity of the findings

-

Additional comments

I was asked just to check code functionality. The Readme added by the authors points to a directory that, unless I'm mistaken, doesn't exist ("Final Project/..."). As such, I was unable to run the code.

Cite this review as
Anonymous Reviewer (2024) Peer Review #2 of "Top-k sentiment analysis over spatio-temporal data (v0.2)". PeerJ Computer Science